# Sit-to-Stand Power Is a Stronger Predictor of Gait Speed than Knee Extension Strength

**DOI:** 10.3390/jfmk9020103

**Published:** 2024-06-13

**Authors:** Garrett M. Steinbrink, Julian Martinez, Ann M. Swartz, Scott J. Strath

**Affiliations:** Zilber College of Public Health, University of Wisconsin-Milwaukee, Milwaukee, WI 53211, USA; steinb83@uwm.edu (G.M.S.); marti994@uwm.edu (J.M.); aswartz@uwm.edu (A.M.S.)

**Keywords:** strength, power, older adults, physical function, screening, mobility, primary care, quality of life

## Abstract

With a growing aging population, the routine assessment of physical function may become a critical component of clinical practice. The purpose of this cross-sectional study is to compare two common assessments of muscular function: (1) isometric knee extension strength (KES) and (2) sit-to-stand (STS) muscle power tests, in predicting objective physical function (i.e., gait speed) in aging adults. 84 adults (56% female, mean (SD) age = 66.6 (9.4) years) had their relative KES, STS power, usual gait speed (UGS), and fast gait speed (FGS) assessed. Multiple linear regression examined the associations between KES, STS power, and gait outcomes. When entered in separate models, KES and STS power were both independently associated with UGS and FGS (Std. β = 0.35–0.44 and 0.42–0.55 for KES and STS power, respectively). When entered in the same model, STS power was associated with UGS and FGS (Std. β = 0.37 [95%CI: 0.15, 0.58] and 0.51 [95%CI: 0.31, 0.70], respectively), while KES was only associated with FGS (Std. β = 0.25 [95%CI: 0.02, 0.48]). STS power seems to be a valid indicator of function in aging adults. Its feasibility as a screening tool for “low” function in the primary care setting should be explored.

## 1. Introduction

Gait speed, or the “functional vital sign” [1], is a powerful predictor of mobility limitations and disability [2], and generally declines during the aging process, likely due to preferential type II muscle fiber denervation and atrophy [3]. Consequently, usual gait speed (UGS) significantly predicts mortality, falls, and hospitalizations in aging adults [4,5,6]. Furthermore, “slow” UGS and self-reported difficulties during walking may result in significant social participation limitations, isolation, and loneliness [7], which may explain the positive relationship between physical function and quality of life in aging adults [8]. As the number of adults over 65 years old in the U.S. is expected to grow from ~60 million to 82 million by the year 2050, the maintenance of gait and physical function will be paramount for the maintenance of health and well-being in aging adults.

Muscle strength, and particularly the maximal strength of the knee extensors, is a strong predictor of physical function and mortality in aging adults [9,10,11]. With respect to gait, longitudinal reductions in knee extension strength (KES) predict declines in gait speed, independent of changes in body composition [12]. Improvements in strength through resistance training interventions elicit positive changes in gait speed and mobility outcomes [13,14], suggesting a clear causal link between strength and physical function in aging adults. As such, in their most recent consensus statement, the European Working Group on Sarcopenia in Older People (EWGSOP) proposed that muscle strength, due to its reliability in assessing skeletal muscle health, be the primary parameter of sarcopenia, or the age-related reduction in skeletal muscle function [15]. While the EWGSOP consensus statement currently suggests grip strength as the primary assessment tool for strength, grip strength is a weaker predictor of functional outcomes compared to lower-extremity strength measures [9,16]. While the clinical utility of grip strength should not be understated, the KES test is a commonly used measure of strength assessment in clinical settings, suggesting it may be as feasible measure of strength in aging adults [17]. Indeed, the isometric KES test is a valid and reliable tool for the assessment of KES [18] and for the screening of general lower-limb function in aging adults [19].

While strength appears to be an important determinant of skeletal muscle function in aging, muscle power, due to its more precipitous age-related decline, has garnered significant attention as a critical indicator of physical functioning in recent years [20]. Indeed, muscle power is a strong predictor of all-cause mortality, physical function, and mobility outcomes in aging adults [20,21,22,23]. Given that power training enhances voluntary skeletal muscle activation [24,25], which is longitudinally associated with improvements in gait speed [26], it seems clear there is a strong, causal link between muscle power and gait performance/mobility. Indeed, similar to strength, resistance exercise-induced increases in lower-extremity power are associated with clinically meaningful changes in UGS [23], and power training is effective at increasing gait speed in both healthy and mobility-limited populations [27,28]. Altogether, these findings support power as an increasingly important indicator of physical function in aging populations, so its routine assessment in aging adults may be warranted.

While both lower-extremity strength and power are seemingly important musculoskeletal determinants of gait speed, investigating the relative importance of strength and power to mobility outcomes is a clinically important question. Interestingly, the maximal leg press power of mobility-limited adults better explains both UGS and fast gait speed (FGS) compared to maximal leg press strength [22]. Additionally, changes in lower-extremity power, but not changes in strength, are associated with clinically meaningful improvements in gait performance [23], and power-based exercise training interventions may be more effective than traditional, strength-based interventions in improving physical function in aging adults [29]. As such, some researchers have suggested that muscle power be treated independent of muscle strength and muscle size due to its greater sensitivity to detect age- and disease-related declines in physical function [30].

One of the issues researchers and practitioners face in assessing muscle power is the lack of valid, safe, and easy-to-use assessment tools [31]. The most frequently used method to assess muscle power in aging adults is the pneumatic leg press [31], which is a relatively expensive, stationary, computer-interfaced resistance machine. To make the assessment of lower-extremity power more clinically feasible, multiple equations have been developed to estimate power during the sit-to-stand (STS) test [32]. Of the equations developed, the equation from Alcazar and colleagues [33] is the strongest predictor of frailty and functional limitations in those 65+ years of age [32]. Furthermore, STS muscle power, estimated from this equation, is associated with physical independence, cognitive function, frailty, and health-related quality of life in aging populations [33,34,35]. Given that both the isometric KES and STS power tests are neither resource- nor time-intensive assessments, they could conceivably be implemented in primary care settings for the routine assessment of age-related functional decline. While STS power is a stronger predictor of falls in aging adults compared to isometric KES [36], their comparative ability to predict objective physical function outcomes (i.e., gait speed/mobility) is not completely understood. Therefore, the purpose of this study is to compare the independent relationships between isometric KES and STS power with respect to UGS and FGS in a sample of aging adults. We hypothesize that STS power will be a stronger predictor of gait speed/mobility compared to KES. The results of this study may influence future functional screening practices for aging adults.

## 2. Materials and Methods

### 2.1. Participants

Participants in the greater Milwaukee, WI, USA metropolitan area were recruited to participate in this study via flyers in common community settings (e.g., community centers, health and fitness facilities, etc.), a laboratory-maintained research participant database, and word-of-mouth. Participants were included in this study if they were community dwellers and were at least 50 years of age. Participants were excluded from the study if they relied on an assistive device, had a current or previous history of cognitive impairment, or were unable to safely follow the study’s procedures. Figure 1 shows a flowchart of the number of included and excluded participants, while Table 1 shows participant characteristics. All participants read the study’s informed consent document and provided their written informed consent prior to completing the study procedures. The study’s procedures were approved by the university’s Institutional Review Board.

Participants visited the laboratory once, with all administered tests being conducted by trained research assistants. Body mass and height were measured by a calibrated physician’s scale and stadiometer, respectively, and BMI was calculated as the ratio of the participant’s body mass to height, squared (kg/m^2^).

### 2.2. Protocol Measurements

Maximal isometric KES was measured unilaterally with a linear force transducer (LCM300, Futek Advanced Sensor Technology, Irvine, CA, USA). After being appropriately situated, with a strap fastened around the participant’s ankle (Figure 2), participants performed one practice trial to familiarize themselves with the assessment. After the practice trial, participants were encouraged to contract their knee extensor muscles (i.e., quadriceps) maximally against the device for a total of 3–5 s for each trial. Each trial was then followed by at least 15 s of rest. Participants completed three trials on both the right and left legs. The maximal force elicited on either leg, irrespective of leg dominance, was normalized to the participant’s body mass for the final analysis. Data were acquired at 1000 Hz using Spike2 software (v6, CED Data Acquisition & Analysis, Cambridge, UK) and filtered using a 0.01-s smoothing filter [37].

Relative STS power was estimated from the 5-repetition STS test (5STS) using a previously validated equation [33,38]. Briefly, participants were asked to perform five consecutive STS transitions as quickly as they could, from a standard-height chair and without using their arms, by crossing them across their chest or abdominal region. The total time taken to complete the test was recorded with a manual stopwatch, and the participant’s body mass, height, time taken to complete the 5STS test, and chair height were entered into the following equation, where body mass is the participant’s body mass in kilograms (kg), the coefficient 0.9 represents the assumed 90% of the participant’s body mass displaced during a STS transition [33], *g* is the acceleration due to gravity (i.e., 9.81 m/s^2^), Height is the height of the participant in meters (m), the coefficient 0.5 represents the assumed proportion of the legs relative to the participant’s body height, Chair height is the height of the chair (i.e., 0.47 m), 5 STS time is the time, in seconds, it took for the participant to complete the 5 STS test, 5 is the number of STS transitions during the 5 STS test, and the coefficient 0.5 represents the assumed proportion of time for 1 concentric portion of a single STS transition. Absolute STS power was then normalized to the participant’s body mass (kg) for final analysis. The equations used to estimate STS power are shown below:
$$Absolute\;STS\;mean\;power\left(W\right)=\frac{Body\;mass\times 0.9\times g\times \left[Height\times 0.5-Chair\;height\right]}{\frac{5\;STS\;time}{5}\times 0.5}$$
$$Relative\;STS\;mean\;power\left(\frac{W}{kg}\right)=\frac{Absolute\;STS\;mean\;power}{Body\;mass}$$

For the assessment of UGS and FGS, participants walked 10 m at their usual and fast walking pace, respectively. To account for acceleration and deceleration at the beginning and end of the 10-m-long track, only the intermediate 6 m were timed with a manual stopwatch. Three trials were conducted for each gait speed outcome, and the average of the trials was used for the final analysis.

### 2.3. Statistical Analysis

Continuous and categorical variables are reported as means and standard deviations (SD) and frequencies (%), respectively. The normality of the data was confirmed with the Shapiro-Wilk test. The unadjusted relationships between KES, STS power, and gait speed outcomes were assessed with Pearson product-moment correlation coefficients (*r*). The adjusted relationships between the KES, STS power, and gait speed were estimated with multiple linear regression analyses. In the first set of models, KES and STS power were entered into the models separately. In the second set of models, KES and STS power were entered in the same model. All analyses were complete-case analyses. If participant data were missing for any exposure, outcome, or covariate, they were excluded from the analysis. Covariates were selected based on their unadjusted relationships with the primary dependent variables, along with previously published data [22]. More specifically, age and BMI, as continuous variables, and sex, as a dichotomous variable (i.e., male or female), were included as covariates in our adjusted models due to their known relationships to strength, power, and physical function. Multi-collinearity between all independent variables in the models was assessed with variation inflation factors (VIFs), and all variables had VIFs < 2, which suggest limited multicollinearity. Standardized beta coefficients (Std. β) and their 95% CIs are reported to compare the strengths of the relationships between KES and STS power and gait outcomes. All statistical analyses were conducted in R Statistical Software (v4.2.1; R Core Team, 2022, Vienna, Austria).

## 3. Results

### Associations between KES, STS Power, and Gait Speed Outcomes

In the unadjusted analyses, compared to KES, STS power was more strongly related to UGS (*r* = 0.48 and 0.52, respectively; both *p* < 0.001) and FGS (*r* = 0.53 and 0.63, respectively; both *p* < 0.001). The regression analyses examining the independent relationships between gait speed outcomes and KES and STS power are shown in Table 2 and Table 3, respectively. KES was significantly associated with UGS and FGS, controlling for age, sex, and BMI (Std. β = 0.35 [95%CI: 0.09, 0.62] and 0.44 [95%CI: 0.19, 0.70], respectively). Similarly, STS power was significantly associated with UGS and FGS, controlling for the same confounders (Std. β = 0.42 [95%CI: 0.21, 0.62] and 0.55 [95%CI: 0.36, 0.74], respectively).

The regression analysis with KES and STS power entered in the same model is shown in Table 4. Controlling for age, sex, BMI, and STS power, KES was not significantly associated with UGS, but was associated with FGS (Std. β = 0.22 [95%CI: −0.04, 0.48] and 0.25 [95%CI: 0.02, 0.48], respectively). Conversely, controlling for age, sex, BMI, and KES, STS power was associated with both UGS and FGS (Std. β = 0.37 [95%CI: 0.15, 0.58] and 0.51 [95%CI: 0.31, 0.70], respectively).

## 4. Discussion

The purpose of this study was to compare the independent relationships between maximal isometric KES and STS power with respect to gait speed outcomes in a sample of aging adults. We found that KES and STS power are both robustly associated with gait speed outcomes in aging adults. Importantly, STS power is more strongly related to gait speed compared to KES. Therefore, the STS power test may be a more viable assessment, compared to KES, for the routine screening of skeletal muscle function in the aging population. These findings may have implications regarding the clinical identification and management of aging and disease-related changes in physical function.

In this study, we found that KES is a strong predictor of gait speed/mobility in aging adults. More specifically, we found that a 1-SD greater isometric KES was associated with a 0.35 and 0.44-SD greater UGS and FGS, respectively. Practically, the magnitude of this relationship would be equivalent to a ~0.13 m/s greater UGS with a 1-SD higher KES, which is largely considered a “clinically meaningful difference” [23]. This finding is consistent with the existing literature. For example, when examining the relationship between KES and gait speed, Fragala and colleagues concluded that having low isometric KES increases the risk of having a slow gait speed (i.e., <0.8 m/s) 3-fold [9]. Similarly, Bean et al. report that maximal leg press strength, irrespective of age, BMI, and chronic disease status, significantly explains both UGS and FGS performance [22]. Contrary to the results of the current and previously published studies, some evidence suggests strength at the ankle joint (i.e., dorsiflexion and plantar flexion strength) may be more related to gait performance [39], compared to KES. Indeed, Kanayama et al. recently showed that ankle plantar flexion strength is significantly associated with maximal gait speed, independent of knee extension velocity and strength, which were largely not associated with gait when plantar flexion strength was in the model [39]. This finding suggests that the demands of the knee extensors during walking are low in aging adults, which is corroborated by evidence indicating a substantial KES “reserve” during walking tasks in this population [40]. Therefore, while we and others have found KES to be significantly associated with UGS and FGS in aging adults, it may not be the best indicator of gait speed and mobility outcomes in this population. Clinical assessments that require the activation of multiple muscle groups and joint actions, therefore, may be more indicative of function in the aging population.

Indeed, similar to muscle strength, this study found that STS power was independently associated with gait speed outcomes in aging adults. Specifically, we found that a 1-SD greater STS power was associated with a 0.42 and 0.55-SD higher UGS and FGS, respectively, which were stronger than the relationships between KES and gait speed outcomes. Overall, these findings are consistent with previous literature. For example, Bean and colleagues were one of the first groups to show that lower-extremity muscle power was an important predictor of gait speed and mobility outcomes in aging adults [22,23,41]. Since then, skeletal muscle power has received considerable attention as an important determinant of physical function in this population [20,30]. With respect to power measured during the STS test, the results of this study are consistent with evidence showing that STS power is a significant predictor of gait speed and mobility outcomes in aging adults. For example, in a sample of 3689 Columbian adults, Ramírez-Vélez et al. reported that having low age- and sex-specific relative STS power was associated with a 2.5–4-fold greater risk of having a gait speed <0.8 m/s [42]. In a similarly designed study, Losa-Reyna and colleagues found that having low relative STS power was associated with a greater likelihood of having low UGS, frailty, and a poor quality of life in both men and women [43]. These previous studies, in addition to the current study, illustrate the utility of the STS power test as a means of assessing physical health-related outcomes (e.g., gait speed, frailty, etc.) in large cohorts of aging adults, and therefore it could be considered an appropriate tool for routine functional screening in this population.

Most importantly, we found that STS power is more strongly related to objectively measured gait speed outcomes compared to KES. Specifically, we found that STS power was strongly related to both gait speed outcomes assessed, independent of KES, while KES was not related to both gait speed outcomes, independent of STS power. This suggests that while KES was significantly related to gait speed on its own, it did not significantly explain any additional variation in these outcomes when STS power was accounted for. Additionally, in all models, STS power showcased higher standardized beta coefficients compared to KES, illustrating its superiority in explaining the gait speed performance of this sample. Overall, these findings are consistent with previous studies. Indeed, Zanker et al. recently showed, in a sample of over 1300 men and women, that relative STS power predicts low usual walking speed (i.e., <0.8 m/s) better than isometric KES [11]. Additionally, Simpkins and Yang concluded that STS power is a stronger differentiator between fallers and non-fallers, compared to KES, in a sample of healthy older adults [36]. Altogether, these previous data and the current study suggest that the STS power test is more indicative of functional outcomes compared to the KES. Given that STS power is also more strongly related to physical independence, falls, and fractures compared to handgrip strength [35,44], the STS power test is a highly applicable functional screening tool in aging adults, and its use may be recommended over these other common assessments of muscular function.

As with any study, this one has both strengths and limitations. The main strength of this study is its direct comparison between two commonly used skeletal muscle function assessment tools for the purpose of predicting objective functional outcomes. As such, the results from this study may better inform the selection of a functional screening tool(s) in the routine care of aging adults. One limitation of this study is its relatively small and generally well-physically functioning sample. Despite the limitations of this sample, recent evidence in a large sample of adults, with a third of these participants having “slow” UGS (i.e., <0.8 m/s), largely agrees with our results and suggests that STS power better predicts slow gait compared to KES [11]. Also, given that maximal walking speed/FGS is more indicative of functional fitness in community-dwelling adults, compared to UGS [45], our findings suggesting that STS power is more strongly related to FGS compared to KES lend additional support for the use of the STS power test to identify changes in physical function in this population. Additionally, while we did find that equation-estimated STS power was a stronger predictor of gait speed outcomes compared to KES, it is unclear whether objectively measuring STS velocity and power with mobile, easy-to-use technologies such as wearable sensors or linear velocity/position transducers would show stronger associations between STS power and functional outcomes, compared to estimation equations. Given that the equation we utilized for this study underestimates STS velocity by ~25% [46], this question is worthy of future investigation.

## 5. Conclusions

STS power, estimated from the frequently used 5STS test, is more strongly related to objective gait speed outcomes compared to the isometric KES test. These findings may have important implications with respect to the routine screening of muscular function during aging and disease processes. Given that exercise intervention, and particularly resistance exercise, is one of the few available therapies for slowing age- and disease-related muscular function decline, confirming and implementing current STS power diagnostic cut-offs as a method of identifying and referring patients to future exercise interventions could help to improve the function and quality of life of the expanding aging population.

## Figures and Tables

**Figure 1 jfmk-09-00103-f001:**
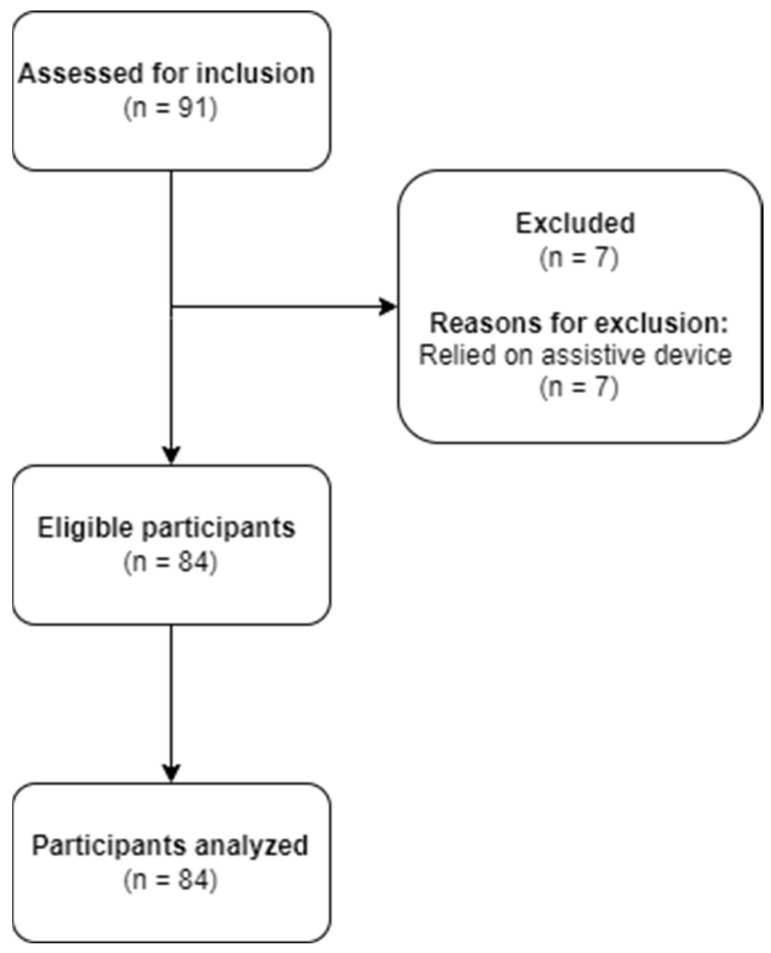
Participant inclusion/exclusion flowchart.

**Figure 2 jfmk-09-00103-f002:**
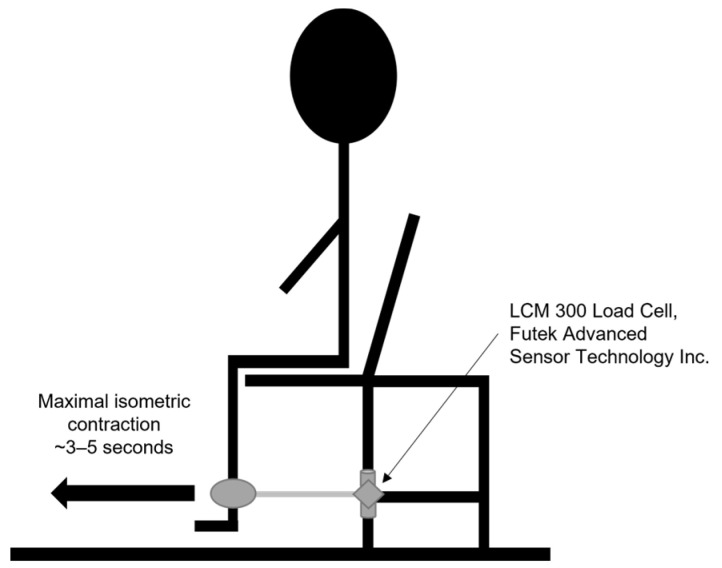
Set up for collecting isometric knee extension strength (KES).

**Table 1 jfmk-09-00103-t001:** Participant characteristics.

Characteristic	n = 84 ^1^
Age, years	66.6 (9.4)
Female	47 (56%)
Race/ethnicity	
American Indian or Alaska Native	1 (1.2%)
Asian	1 (1.2%)
Black	5 (6.0%)
White	74 (88%)
Hispanic	1 (1.2%)
Other	2 (2.4%)
Highest education completed	
High school	14 (17%)
College	40 (48%)
Graduate school	30 (36%)
Annual household income	
<USD 5000	1 (1.4%)
USD 5000–14,999	3 (4.2%)
USD 15,000–24,999	9 (13%)
USD 25,000–34,000	12 (17%)
USD 35,000–49,999	11 (15%)
>USD 50,000	36 (50%)
Did not provide	12
BMI, kg/m^2^	28.1 (5.7)
Relative knee extension strength, kg/kg	0.33 (0.10)
Relative STS power, W/kg	2.69 (0.97)
Usual gait speed, m/s	1.27 (0.35)
Fast gait speed, m/s	1.76 (0.65)

^1^ Mean (SD); n (%).

**Table 2 jfmk-09-00103-t002:** Examining the associations between relative knee extension strength (KES), usual gait speed (UGS), and fast gait speed (FGS), controlling for age, sex, and BMI.

	UGS			FGS		
Characteristic	Std. Beta	95% CI	*p*-Value	Std. Beta	95% CI	*p*-Value
Age	0.01	−0.21, 0.22	0.95	−0.01	−0.22, 0.19	0.91
Sex (Male)	0.02	−0.21, 0.24	0.89	0.01	−0.21, 0.22	0.96
BMI	**−0.26**	**−0.49, −0.03**	**0.03**	−0.18	−0.40, 0.05	0.12
Relative KES	**0.35**	**0.09, 0.62**	**0.01**	**0.44**	**0.19, 0.70**	**<0.001**

Std. Beta = standardized beta coefficient; 95% CI = 95% confidence interval. Bolded coefficients, 95% CIs, and *p*-values are statistically significant at an alpha level = 0.05.

**Table 3 jfmk-09-00103-t003:** Examining the associations between relative sit-to-stand (STS) power, usual gait speed (UGS), and fast gait speed (FGS), controlling for age, sex, and BMI.

	UGS			FGS		
Characteristic	Std. Beta	95% CI	*p*-Value	Std. Beta	95% CI	*p*-Value
Age	0.03	−0.17, 0.22	0.78	0.00	−0.18, 0.17	0.96
Sex (Male)	0.08	−0.12, 0.27	0.44	0.07	−0.11, 0.25	0.42
BMI	**−0.28**	**−0.48, −0.08**	**0.01**	**−0.19**	**−0.38, −0.01**	**0.04**
Relative STS power	**0.42**	**0.21, 0.62**	**<0.001**	**0.55**	**0.36, 0.74**	**<0.001**

Std. Beta = standardized beta coefficient; 95% CI = 95% confidence interval. Bolded coefficients, 95% CIs, and *p*-values are statistically significant at an alpha level = 0.05.

**Table 4 jfmk-09-00103-t004:** Examining the associations between relative knee extension strength (KES), relative sit-to-stand (STS) power, usual gait speed (UGS), and fast gait speed (FGS), controlling for age, sex, and BMI, with KES and STS power in the same model.

	UGS			FGS		
Characteristic	Std. Beta	95% CI	*p*-Value	Std. Beta	95% CI	*p*-Value
Age	0.04	−0.16, 0.24	0.68	0.03	−0.15, 0.21	0.75
Sex (Male)	−0.02	−0.23, 0.19	0.87	−0.04	−0.23, 0.15	0.65
BMI	−0.20	−0.42, 0.02	0.07	−0.10	−0.30, 0.10	0.31
Relative KES	0.22	−0.04, 0.48	0.09	**0.25**	**0.02, 0.48**	**0.03**
Relative STS power	**0.37**	**0.15, 0.58**	**0.001**	**0.51**	**0.31, 0.70**	**<0.001**

Std. Beta = standardized beta coefficient; 95% CI = 95% confidence interval. Bolded coefficients, 95% CIs, and *p*-values are statistically significant at an alpha level = 0.05.

## Data Availability

Study data will be made available upon reasonable request.

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
