# Peer review of "Sit-to-Stand Power Is a Stronger Predictor of Gait Speed than Knee Extension Strength"

_jfmk, 2024, doi:10.3390/jfmk9020103_

Round 1

Reviewer 1 Report

Comments and Suggestions for Authors

Review jfmk-3037040-peer-review-v1

The paper Sit-to-stand Power is a Stronger Predictor of Gait Speed than Knee Extension Strength aims to compare two commonly used assessments of muscular function- namely, isometric knee extension strength (KES) and sit-to-stand (STS) muscle power tests- in predicting objective function (i.e., gait speed) in aging adults. Overall, the paper is well-written and clear. My comments pertain to the "Materials and Methods" and "Results" sections. Below are the elements that the authors add.

1.     Material and Methods

Please, provide exclusion criteria for participants.

Lines 109 - 116 - please include a figure showing how the individuals were blocked and in what position they were sitting.

Line 114 - 115 - what was taken for analysis, the average of three trials for one leg, or the maximum value of three trials for one leg, maybe for two legs? Please elaborate, as it is not written.

Line 124 - Please provide explanations for the variables in the equation. What is W, g, Height - which one? 5STStime - what does it mean?

Line 119 - in this case, where were the hands placed: on the hips, on the shoulders?

Line 130 - Please write exactly the mean (SD), and what is SD (of course it is known, but the symbol appears for the first time).

Was the normality of the distribution of the data checked, and with what test? Please add information.

Line 132 - Pearson correlation - then "r" appears in the results, so this symbol needs to be introduced in Material and Methods.

Line 139 - what are variation inflation factors (VIFs)? What formula was it calculated with?

Line 140 - what is Standardized beta coefficients (β)? How was it counted? Why does it appear later in the results section, for example, in Table 2 Std. Beta? Is it the same thing? Please standardize the labels.

2.     Results

Line 151 - did the results given were statistically significant?

I don't see the results for UGS and FGS, were they statistically significant? Please provide this information in the text, preferably at the beginning of the results section. Additionally, Table 1 should appear on line 148.

Table 2, 3, 4 - please correct the following. Please introduce UGS and FGS with the names of Usual gait speed (in Table 2 written with a hyphen and in Table 3 without, please unify) and Fast gait speed. Please specify what * means. What do the two variables mean for 95% CI? Std. Beta - please standardize the labels with those in the Materials and Methods section. Under the tables, it is written: Bolded and italicized coefficients are statistically significant at an alpha level = 0.05, yet nothing is italicized, what does this mean? Please complete the tebels with p-values.

Reviewer 2 Report

Comments and Suggestions for Authors

The purpose of this study is to compare the independent relationships be-92 tween isometric KES and STS power, with respect to UGS and FGS, in a sample of aging adults.

L98, add STROBE checklist for observational study.

L136, describe the Covariates clearly.

L147, describe patients flow. How many patients did you exclude?

Round 2

Reviewer 1 Report

Comments and Suggestions for Authors

Dear Authors,

The paper has been greatly improved and is much clearer. However, I still have a few comments to make it more readable and clear. These are just small corrections:

  1. Captions should be under the figures and above the tables. Currently, every figure is incorrectly labeled. Please adjust accordingly.
  2. Abstract:
    • Line 12: As the mean age is provided, please include the standard deviation and unit.
    • Line 13: The abbreviation "UGS" is not defined. For "FGS" it is fine.
  3. Section 2: Please divide according to the following key:
    • Line 96: 2.1. Participants
    • Line 116: 2.2. Protocol Measurement
    • Line 161: 2.3. Statistical Analysis
  4. Section 3.1. should be removed (the Participants' characteristics are already detailed in the Methods section).
  5. In Tables 2 and 3, please include the abbreviations "UGS" and "FGS" in the first lines.
    • Table 1 should be moved to line 115.
